# The Scandinavian Embedding Benchmarks: Evaluating Multilingual and Monolingual Text Embedding for Scandinavian languages

**Kenneth Enevoldsen**
Aarhus University
kenneth.enevoldsen@cas.au.dk

**Márton Kardos**
Aarhus University
martonkardos@cas.au.dk

**Niklas Muennighoff**
n.muennighoff@gmail.com

**Kristoffer Laigaard Nielbo**
Aarhus University
kln@cas.au.dk

## Abstract

The evaluation of English text embeddings has transitioned from evaluating a handful of datasets to broad coverage across many tasks through benchmarks such as MTEB. However, this is not the case for multilingual text embeddings due to a lack of available benchmarks. To address this problem, we introduce the Scandinavian Embedding Benchmark (SEB). SEB is a framework that enables text embedding evaluation for Scandinavian languages across 24 tasks, 10 subtasks, and 4 task categories. Building on SEB, we evaluate more than 26 models, uncovering significant performance disparities between public and commercial solutions not previously captured by MTEB. We open-source SEB[1] and integrate it with MTEB, thus bridging the text embedding evaluation gap for Scandinavian languages.

## 1 Introduction

Natural language embeddings are used in a diverse range of applications, including clustering (Liu and Xiong, 2011; Angelov, 2020), text mining (Jiang et al., 2015), semantic search (Reimers and Gurevych, 2019a; Muennighoff, 2022) and feature representation (Alayrac et al., 2022). Furthermore, embeddings are crucial in retrieval augmented generation (RAG) systems (Borgeaud et al., 2022), particularly for low- to mid-resource languages and domains. RAG systems enable the enrichment of generative models with the knowledge that might be underrepresented or absent during training. Thus, they can play a role in broadening linguistic and domain coverage.

With the breadth of applications for text embeddings, a proper evaluation of their quality is critical. Recent work has proposed Massive Text Embedding Benchmark (MTEB) (Muennighoff et al., 2023), a benchmark for evaluating the quality of document embeddings for a wide variety of tasks. MTEB improves upon prior benchmarks by addressing the lack of evaluations across tasks. This has led to the widespread adoption of the benchmark for evaluating natural language embeddings.

However, while MTEB substantially improves the evaluation of text embeddings, the benchmark has the following shortcomings:

1. **Support for non-English evaluation:** MTEB contains only limited support for evaluating non-English embeddings and multiple task categories are predominantly covered by

---

[1]https://github.com/KennethEnevoldsen/scandinavian-embedding-benchmark

38th Conference on Neural Information Processing Systems (NeurIPS 2024) Track on Datasets and Benchmarks.

translated datasets (classification) and important task such as retrieval has no multilingual support.

2. **Reproducibilty:** MTEB does not include model implementations in the benchmark's code[2]. This is especially problematic since recent approaches such as prompt-based embedding models (Muennighoff, 2022; Xiao et al., 2023; Su et al., 2023), Matryoshka embeddings (Kusupati et al., 2022) introduce variables which can dramatically influence performance.

3. **Coverage:** While MTEB has broad coverage across tasks, its domain coverage is still limited, as it primarily includes datasets from academic articles, social media, and web sources. This lack of coverage is especially pronounced for non-English tasks.

Our work is driven by the reality that Scandinavian research, public institutions, and industry have to make decisions about their choice of text embedding model for various use cases. These choices are currently made in the absence of a reliable standard to evaluate text embedding models' performance on Scandinavian languages. As a result, these institutions have relied on proxies, such as models' performance on predominantly English benchmarks or Bitext mining tasks. As we demonstrate, performance on these tasks is not necessarily transferable to Scandinavian applications, thus not properly accounting for these institutions' requirements. By introducing a benchmark tailored for Scandinavian languages, we aim to aid these organizations in making informed decisions. Additionally, SEB will presumably support the development of Scandinavian embedding models by providing a standardized means for evaluating new models and comparing them against previously existing ones.

## 1.1 Contributions

To mitigate these issues, we present SEB a benchmark for embedding evaluation of the Mainland Scandinavian languages: Danish (da), Swedish (sv), and Norwegian (Bokmål (nb) and Nynorsk (nn)) as well as the Danish dialect Bornholmsk (da-bornholm). Due to the limited resources available for these languages we choose to utilize the substantial cross-lingual transfer between these languages demonstrated by Nielsen (2023); this supports collectively benchmarking the Mainland Scandinavian languages to broaden the coverage otherwise limited for these languages.[3] SEB makes the following main contributions; (1) it greatly expands the evaluation of embedding for Scandinavian to multiple tasks (see Table 1b) as well as across a wide range of domains (see Table 1a); (2) SEB implements a model registry that allows for the easy addition of new models as well as documents the exact implementation of existing models evaluated in the benchmark. Lastly, (3) SEB expands and extends MTEB by porting all tasks, allowing for the expansion of MTEB to a fully-fledged multilingual benchmark for embeddings. Using SEB we evaluate 26 representative models and APIs within this work and present additional models in an interactive online dashboard.[4]

## 2 Related Work

### 2.1 Benchmarks

Benchmarks are important tools for model development that enable the assessment of significant performance improvements. Prior benchmarks for evaluating text embeddings focused on specific embedding qualities; BEIR (Thakur et al., 2021) and MIRACL (Zhang et al., 2023) assessed embedding efficacy in information retrieval across diverse domains or languages, while SentEval (Conneau and Kiela, 2018) integrated various SemEval datasets for sentence encoding evaluation using semantic text similarity (STS) tasks. MTEB (Muennighoff et al., 2023) amalgamated and expanded these methodologies to cover eight different tasks. While MTEB includes more than 112 languages, most of this linguistic variation originates from only a handful of tasks, notably bitext mining (Tatoeba Project Contributors, 2023) or translated datasets (FitzGerald et al., 2023). Scandinavian languages

---

[2]This can, for instance, be seen in issues such as `https://github.com/embeddings-benchmark/mteb/issues/109`

[3]Please note that the Insular Scandinavian languages, Icelandic and Faroese, are not included as previous research (Nielsen, 2023) has demonstrated only a limited degree of cross-lingual transfer between the Mainland and Insular Scandinavian languages.

[4]`https://kennethenevoldsen.github.io/scandinavian-embedding-benchmark/`

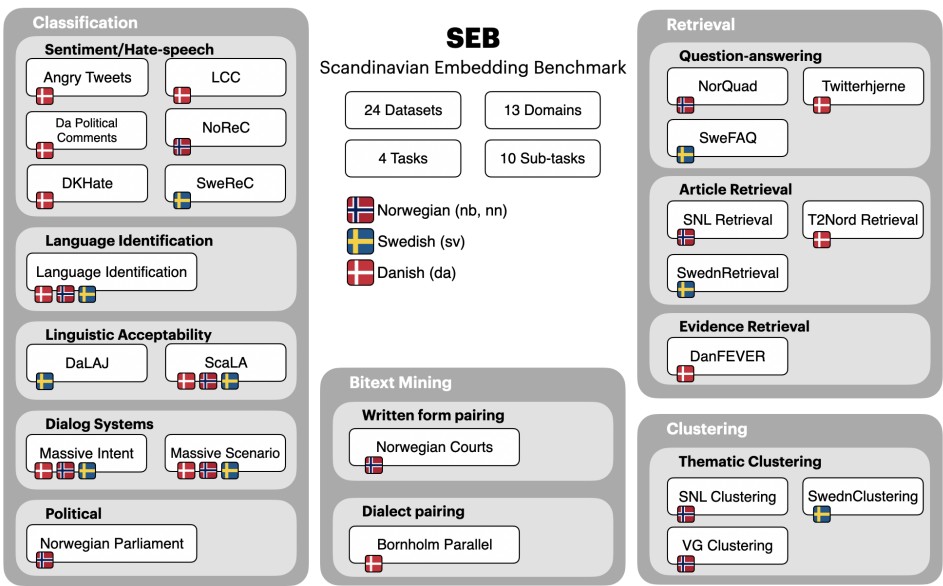

Figure 1: An overview of the tasks and datasets in SEB. Flags denote the languages of the datasets.

are only represented in two datasets for intent and scenario classification (FitzGerald et al., 2023), both of which are translations. Thus, the benchmark contains no naturally occurring text for either of these languages.

While benchmarks for Scandinavian languages have been developed, most – akin to (Super)GLUE (Wang et al., 2018, 2019) – seek to evaluate the performance of multiple natural language understanding tasks. These include monolingual benchmarks such as the Swedish superlim (Berdicevskis et al., 2023), the Norwegian NorBench (Samuel et al., 2023), or cross-lingual benchmarks such as ScandEval (Nielsen, 2023). While these benchmarks are instrumental for developing Scandinavian models, none focus on evaluating text embeddings for, e.g., retrieval or clustering.

## 2.2 Text Embeddings

Over time, the development of dense text embedding models has evolved from focusing on individual words (Mikolov et al., 2013; Pennington et al., 2014) to encompass entire sentences (Conneau et al., 2017; Ni et al., 2022), and currently extends to processing multiple sentences in a wide range of tasks (Xiao et al., 2023; Su et al., 2023; Muennighoff et al., 2024). As is common in natural language processing (Xue et al., 2021), English-centric models have led this development, followed by multilingual models with only a short delay. While multilingual word embedding models already exist (Artetxe and Schwenk, 2019), multitask sentence-level multilingual embedding models are just beginning to emerge (Chen et al., 2024; Wang et al., 2022). However, their progress is hindered by the lack of comprehensive evaluation for multilingual tasks. This evaluation gap hinders progress in the field, preventing us from effectively evaluating model improvements. Our work aims to address this problem to enable further progress and proliferation of multilingual text embeddings.

## 3 The Benchmark

### 3.1 Design and Curation Rationale

SEB seeks to provide an estimate of the quality of embedding for Scandinavian languages and multilingual use cases. To do so, we focus on

**a) Coverage:** The benchmark should cover a wide variety of tasks spanning distinctly different domains, usages, and embedding tasks; SEB compromises 24 datasets spanning at least 12 domains across nine different tasks with broad coverage for each language.

**b) Cultural integrity and model equity:** Recent studies (Berdicevskis et al., 2023; Nielsen, 2023; Muennighoff et al., 2023) have increasingly adopted the strategy of leveraging translated English datasets as a means to evaluate the performance of models in low-resource language contexts. However, we avoid adding such translations, aiming to represent Scandinavian contexts accurately and mitigate the risk of artificially inflating multilingual model capabilities. This decision stems from the recognition that multilingual models, often trained on parallel or translated data (Reimers and Gurevych, 2020), may exhibit inflated performance when evaluated on similarly translated tasks — a hypothesis that, while plausible, remains to be conclusively shown. We choose to keep the existing translated datasets from MTEB within SEB to maintain compatibility.

**c) Cross-lingual generalization:** Given the limited availability of datasets for the Scandinavian languages, we rely on the high degree of cross-lingual transfer (Nielsen, 2023) to estimate model performance more accurately. This approach capitalizes on intrinsic linguistic similarities and shared cultural contexts to bridge data gaps.

**d) Reproducibility and Accessibility:** SEB expands upon the reproducibility of MTEB by including a model registry for all evaluated models to ensure the exact method (e.g., model prompts) for obtaining the results is known. Furthermore, to ensure that the benchmark is as widely accessible as possible, we have limited the size of most datasets to a maximum of 2048 examples. For most models, this allows running the benchmark on a consumer-grade laptop while ensuring proper performance estimation. The benchmark also implements a public cache, allowing users to experiment without needing to rerun models run by others.

In addition to these criteria, SEB follows the desiderata outlined by Muennighoff et al. (2023), allowing for easy extension of the benchmark and providing a simple API and command-line interface making it easy to benchmark models that are not part of SEB by default.

### 3.2 Datasets

We present an overview of the tasks in SEB in Figure 1. Additionally, we have created an overview of the datasets in Table 6, including dataset statistics and a short description of each dataset. Subsection A.4 described the method of evaluation, and Subsection A.5 described the formalization of the specific datasets to the task. SEB seeks to cover a large variety of domains and task types, greatly expanding upon what was previously available for non-English languages within MTEB (see Table 1a and 1b). To allow for the exploration, we add an embedding map of samples from the dataset in Subsection A.3, where it is clearly seen that the datasets occupy different clusters. Similarly, Figure 2 reveals distinctly different clusters of datasets, e.g., the high similarity between SNL Retrieval and NorQuad as both are constructed from encyclopedic sources while distinct datasets such as SweFAQ (Berdicevskis et al., 2023), covering FAQ related to the public sector.

## 4 Methodology

We describe the construction of the datasets in Subsection A.5. To keep our benchmark compatible with MTEB we follow a similar approach for computing scores, these are described in Subsection A.4.

### 4.1 Models

For our benchmarked models, we have chosen a series of representative models seeking to cover a range of model architectures, model sizes, and commercial APIs, as well as models claiming state-of-the-art results on various embedding tasks. In addition, the online dashboard includes additional models not represented here. We group the models into self-supervised and supervised methods.

**Self-supervised methods:**

**Encoders** such as BERT models (Devlin et al., 2019) including monolingual or Scandinavian models trained for Danish (Enevoldsen et al., 2023), Norwegian (Kummervold et al., 2021) and Swedish (Rekathati, 2021) as well as the multilingual model XLM-R (Conneau et al., 2020). We also include a SimCSE (Gao et al., 2021) version of the dfm-encoder-large to indicate the potential performance

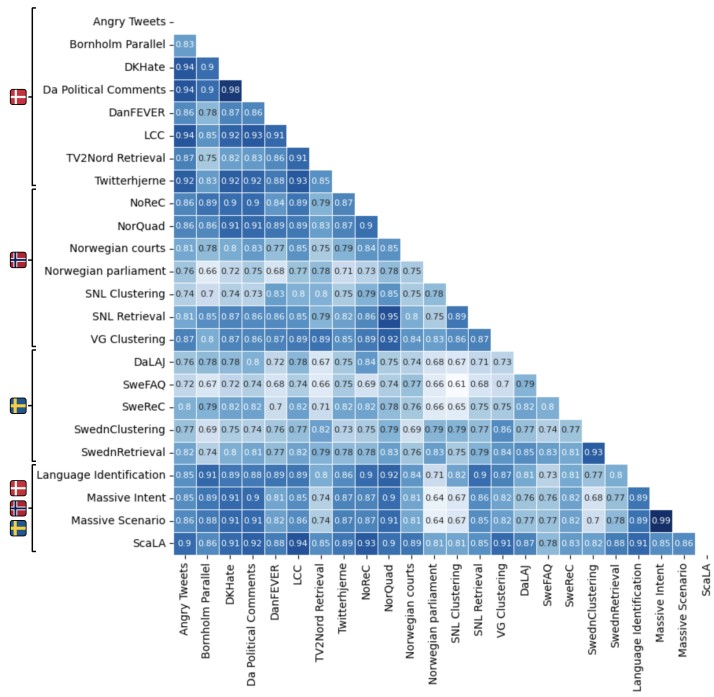

Figure 2: Dataset similarity between the datasets included within SEB. Embeddings are obtained by applying the embed-multilingual-v3.0 on 100 randomly sampled documents. Similarity is computed using cosine similarity.

Table 1: Coverage on Mainland Scandinavian languages. The green plus (+) indicates newly added, while "++" indicates previously not covered in MTEB by any language. The parenthesis is due to the LCC Nielsen (2016) containing the domains, but only to a limited extent. Black checks (✓) indicate domains covered in MTEB for Scandinavian Languages, though only within translated datasets. The domains follow the categorization of the Universal Dependencies Nivre et al. (2017).

(a) Domain Coverage

| Domain | Language | | | |
|---|---|---|---|---|
| | da | nb | nn | sv |
| Academic | (+) | | | |
| Bible | | | | |
| Blog | | | | |
| Fiction | + | + | + | + |
| Government | ++ | ++ | ++ | ++ |
| Legal | (++) | ++ | ++ | |
| Medical | | | | |
| News | + | + | | + |
| Non-fiction | + | + | | + |
| Poetry | (++) | | | |
| Reviews | | + | | |
| Social | + | | | + |
| Spoken | (✓) | (✓) | | (✓) |
| Wiki | + | + | + | + |
| Web | + | | | + |

(b) Task Coverage

| Task | Language | | | |
|---|---|---|---|---|
| | da | nb | nn | sv |
| **Retrieval** | | | | |
| Question answering | + | + | | + |
| Article retrieval | ++ | ++ | | ++ |
| **Bitext Mining** | | | | |
| Dialect pairing | ++ | ++ | ++ | ++ |
| Written form pairing | ++ | ++ | ++ | ++ |
| **Classification** | | | | |
| Political | | ++ | ++ | ++ |
| Language Identification | + | ++ | ++ | ++ |
| Linguistic Acceptability | ++ | ++ | ++ | ++ |
| Sentiment/Hate Speech | + | + | | + |
| Dialog Systems | (✓) | (✓) | (✓) | (✓) |
| **Clustering** | | | | |
| Thematic Clustering | + | + | | + |

Table 2: Performance across task-type categories and languages in SEB. The best score in each model category is highlighted in bold. Additional model evaluation can be found on the public Dashboard. Rank is calculated across all models within the benchmark. The brackets indicate the 95% confidence interval, obtained by bootstrapping 100 repetitions with tasks to minimize the impact of any single task. The symbol "*" signifies when the top-performing model significantly outperforms the second-best model within the same category at a 0.05 significance threshold. Ranks are reported using two significant figures.

| | Avg. rank | Avg. | Task-Type | | | | Language | | | |
|---|---|---|---|---|---|---|---|---|---|---|
| | | | Bitext | Class. | Clust. | Retr. | da | nb | nn | sv |
| Num. Datasets (→) | 24 | 24 | 2 | 12 | 3 | 7 | 12 | 11 | 3 | 9 |
| *Self-Supervised Models* | | | | | | | | | | |
| dfm-encoder-large | 23 (19-26) | 41.4 | 46.8 | 56.5 | 26.9 | 20.1 | 47.7 | 47.4 | 72.5 | 43.7 |
| + SimCSE | 19 (16-22) | **46.6** | 50.9 | 58.4 | 26.9 | **33.7** | **52.2** | 51.3 | 74.3 | 42.0 |
| xlm-roberta-large | 25 (21-30) | 35.3 | 19.1 | 54.6 | 28.1 | 10.0 | 39.6 | 41.3 | 58.0 | 44.5 |
| nb-bert-large | **17** (13-20) | 46.0 | 47.3 | **59.3** | **35.7** | 27.3 | 46.8 | **57.2** | **80.4** | **50.2** |
| nb-bert-base | 21 (18-25) | 42.1 | **51.0** | 57.0 | 31.8 | 18.4 | 43.6 | 53.0 | 79.2 | 47.7 |
| bert-base-swedish | 28 (24-31) | 35.2 | 39.1 | 49.7 | 26.2 | 13.2 | 34.0 | 41.1 | 62.2 | 43.6 |
| fasttext-cc-da | 32 (29-34) | 37.3 | 42.4 | 48.8 | 21.8 | 22.7 | 39.0 | 43.2 | 66.4 | 38.7 |
| fasttext-cc-nn | 32 (29-35) | 35.8 | 47.6 | 46.2 | 22.1 | 20.4 | 34.6 | 43.9 | 69.1 | 37.1 |
| fasttext-cc-nb | 30.0 (27-32) | 37.5 | 43.2 | 48.7 | 24.2 | 22.2 | 37.5 | 45.6 | 67.7 | 38.9 |
| fasttext-cc-sv | 31.5 (29-34) | 36.0 | 43.3 | 47.3 | 22.0 | 20.4 | 34.9 | 41.3 | 63.4 | 40.6 |
| *Supervised Models* | | | | | | | | | | |
| multilingual-MiniLM-L12 | 20 (17-23) | 50.0 | 51.0 | 53.7 | 31.7 | 51.1 | 49.9 | 52.7 | 58.3 | 50.3 |
| multilingual-mpnet-base | 16 (13-20) | 53.2 | 52.7 | 56.5 | 32.7 | 56.5 | 53.0 | 55.8 | 59.6 | 53.3 |
| labSE | 18 (15-21) | 50.5 | 69.1 | 53.6 | 29.0 | 48.9 | 50.9 | 52.9 | 59.4 | 48.7 |
| sentence-bert-swedish | 23 (19-26) | 46.6 | 43.3 | 51.0 | 35.6 | 44.6 | 43.2 | 48.2 | 62.7 | 54.7 |
| e5-mistral-7b-instruct | **8.7** (6.8-12) | 60.4 | **70.8** | 61.7 | 35.7 | 66.0 | **61.7** | 62.9 | 68.8 | 60.4 |
| multilingual-e5-large | 8.8 (6.0-12) | **60.7** | 60.1 | **62.5** | 34.2 | **69.1** | 61.1 | **63.1** | **73.9** | **62.8** |
| multilingual-e5-base | 12 (9.4-15) | 57.9 | 61.4 | 60.1 | 34.0 | 63.5 | 58.6 | 60.9 | 72.0 | 58.5 |
| multilingual-e5-small | 14 (11-16) | 56.4 | 61.6 | 58.1 | **36.9** | 60.3 | 56.5 | 58.9 | 69.5 | 57.1 |
| translate-e5-large | 21 (18-24) | 47.7 | 50.7 | 54.7 | 27.3 | 43.4 | 49.0 | 50.1 | 59.2 | 59.2 |
| sonar-dan | 23 (20-26) | 43.4 | 70.5 | 53.5 | 19.6 | 28.6 | 48.3 | 46.0 | 63.7 | 42.9 |
| sonar-nob | 25 (21-28) | 41.5 | 63.2 | 52.9 | 18.5 | 25.6 | 45.2 | 45.9 | 64.7 | 42.4 |
| sonar-nno | 25 (22-28) | 41.5 | 65.5 | 52.8 | 17.3 | 25.7 | 45.5 | 45.1 | 63.2 | 42.6 |
| sonar-swe | 24 (21-27) | 42.8 | 70.7 | 52.9 | 19.4 | 27.6 | 47.1 | 45.4 | 63.1 | 42.9 |
| *Embedding APIs* | | | | | | | | | | |
| text-embedding-3-large | **5.8** (3.3-8.2) | **65.0** | **68.8** | 63.5 | 38.7 | **77.9** | **63.7** | **69.0** | **74.7** | **65.5** |
| text-embedding-3-small | 9.4 (7.7-12) | 61.0 | 66.7 | 59.7 | 38.3 | 71.3 | 59.7 | 64.7 | 70.2 | 60.4 |
| embed-multilingual-v3.0 | 6.1 (3.8-8.9) | 64.1 | 64.2 | **63.6** | **40.2** | 75.2 | 62.6 | 68.5 | 74.1 | 64.3 |

gain by self-supervised pre-training. This model is trained on sentences extracted from the Danish Gigaword (Strømberg-Derczynski et al., 2021) using default parameters[5].

As a candidate for **Static Word Vectors**, we include four fastText (Joulin et al., 2016, 2017; Bojanowski et al., 2017) models for Danish, Swedish, and Norwegian Bokmål and Nynorsk respectively.

**Supervised Methods:**

For **encoders**, we benchmark LaBSE (Feng et al., 2022), which is based on BERT but further pre-trained on a parallel corpus. Further, we evaluate the multilingual MiniLM models and MPNet models (Reimers and Gurevych, 2019b; Song et al., 2020; Wang et al., 2021), which are trained on diverse datasets. We also include the SONAR models (Duquenne et al., 2023) as they claim improved performance over LabSE. In addition, we include the Swedish sentence transformers (Rekathati, 2021) trained with knowledge distillation from an English model (Reimers and Gurevych, 2020).

---

[5]For exact specification see the model card; `https://huggingface.co/KennethEnevoldsen/dfm-sentence-encoder-large`

Because the development of Scandinavian **decoders** is only in its early stages (Enevoldsen et al., 2023; Ekgren et al., 2022), we utilize the e5-mistral model (Wang et al., 2022, 2023) as it presents a competitive model in the category.

**Commercial embedding APIs:** We additionally include the embedding APIs of Cohere [6] and OpenAI [7] to compare openly available models with commercial solutions.

Lastly, we add **Translate and embed** as a baseline model for comparing naïvely translating to English and then embedding with high-quality English models. To allow for comparison with multilingual models, we include both the large English e5 model and all sizes of its multilingual variants (Wang et al., 2022). We use the multilingual M2M100 model (Fan et al., 2020) for the translation. For translation, we assume the language is known. This avoids accumulating errors due to language detection, and in many applications, the language would be known. We assume Danish as the origin for tasks requiring multiple languages, such as bitext mining.

# 5 Results

In Table 2, we see that the best-performing model is either of the commercial APIs of OpenAI and Cohere followed by the publicly available multilingual e5 model series (Wang et al., 2022). This stands in contrast to developments observed from ScandEval (Nielsen, 2023), where notably smaller monolingual or Scandinavian models have proven to be competitive, often surpassing significantly larger multilingual models. Similar to MTEB (Muennighoff et al., 2023), we find a pronounced performance between self-supervised methods and their supervised counterparts, although we see that notable gains can be obtained from unsupervised pre-training (Gao et al., 2021). In general, however, utilizing unsupervised contrastive pretraining pales in comparison to popular multilingual models of smaller size.

In Table 5, we see the performance across domains. Generally, we see that model rankings remain relatively stable across these domains, with the e5 models (Wang et al., 2022) and the commercial APIs taking a consistent lead. However, we also see that in domains such as the legal domain, spoken language, and fiction, we see the e5-mistral-7b-instruct outcompeting commercial solutions.

If we examine individual subtask (see Subsection A.8) Pretrained encoders perform surprisingly well on language acceptability and language detection tasks. This is likely due to a trade-off between semantics and syntax. Self-supervised training on natural language will likely assign significance to syntactic nuances, while models trained on semantic tasks ignore some syntactical errors favoring semantics.

**Performance across task-types:** Models that have been contrastively trained on sentence pairs or finetuned for a set of common tasks typically outperform pre-trained models, especially in retrieval contexts, while LaBSE (Feng et al., 2022) and the SONAR models (Duquenne et al., 2023), which has been designed for bitext-mining purposes, excels at the task.

The largest gap between commercial and public models is in retrieval, where performance drops more than eight points. While notable improvements have been achieved in publicly available embedding models for English retrieval tasks (Wang et al., 2023), similar results are yet to be achieved in multilingual contexts. Bitext mining is the only category in which open solutions outperform commercial solutions.

**Translate then embed:** When comparing the 'translate-then-embed' model against the multilingual e5 models, we see that in almost all cases, the multilingual models perform better even when comparing across size categories. While performance could likely be improved by utilizing state-of-the-art embedding and translation models, we see few benefits to this approach due to increased computational costs, model complexity, and competitive approaches for knowledge distillation across languages (Reimers and Gurevych, 2020).

---

[6] `https://txt.cohere.com/introducing-embed-v3/`
[7] `https://openai.com/blog/new-embedding-models-and-api-updates`

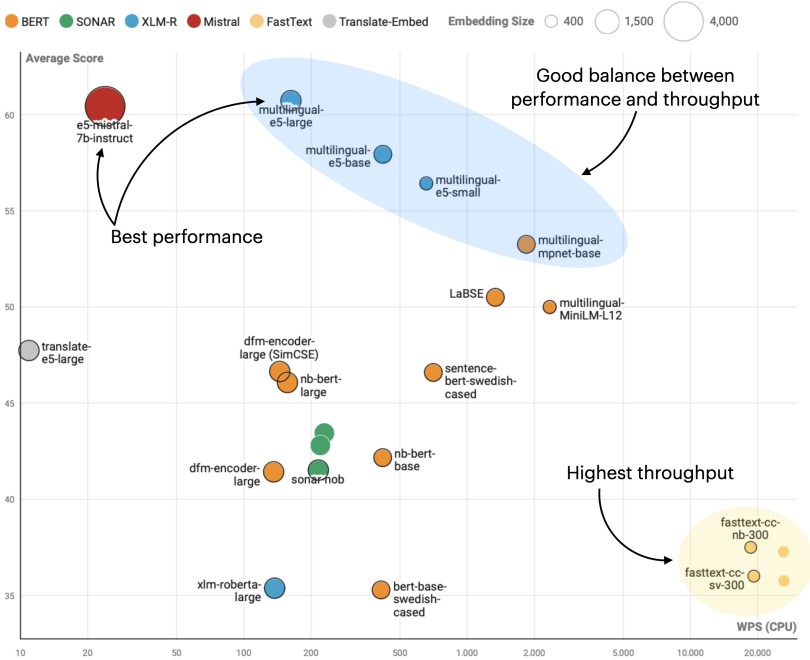

Figure 3: Performance and speed of embeddings models. The size of the circles denotes the embedding size, and the color denotes the model type. Note that commercial APIs are not included. WPS stands for words per second. We avoid highlighting all models to improve readability.

## 5.1 Efficiency

We examine the trade-offs between performance and speed in Figure 3. Speed was benchmarked on Dell PowerEdge C6420 Intel(R) Xeon(R) Gold 6130 CPUs with 32 cores/CPU. We see the following categories of relevance;

**Highest Throughput** FastText models offer the highest throughput while maintaining an average performance exceeding that of the multilingual XLM-R (Conneau et al., 2020).

**Maximum Performance** Achieving optimal performance is possible with the multilingual-e5-large or the e5-mistral-7b-instruct, which rivals the smaller commercial embedding APIs.

**Balanced Performance:** The best balance between performance, throughput, and embedding size is seen in the multilingual e5 models series, which performs competitively on the benchmark. The multilingual-mpnet-base also offers a competitive balance between throughput and performance, despite its larger embedding size.

## 5.2 Comparison with MTEB

As seen in Table 3, where we compare our results with those of MTEB we see that English-focused models (e5-size) perform significantly worse on SEB, while we see the inverse for multilingual models. The e5-mistral-7b-instruct model, based on the multilingual Mistral 7b Jiang et al. (2023), shows notably poorer performance on non-English data, likely due to Mistral's not being trained on Scandinavian languages. While English has several open-source models that perform on par with APIs, this trend is not as evident for Scandinavian languages. Among multilingual models, MTEB and SEB rankings generally align, though selecting models for Scandinavian languages based solely on MTEB results could lead to suboptimal choices (e.g., e5-mistral-7b-instruct, LaBSE).

## 5.3 Limitations and Future Perspectives

**Domain Coverage**: Despite the advancements introduced by SEB, the benchmark could further benefit from encompassing domains crucial to the welfare states of Scandinavia, such as legal,

| Model | SEB Rank (Avg.) | MTEB Rank (Avg.) | Difference Rank |
|---|---|---|---|
| text-embedding-3-large | 1 (65.0) | 3 (64.6) | +2 |
| embed-multilingual-v3.0 | 2 (64.1) | 4 (64.0) | +2 |
| text-embedding-3-small | 3 (61.0) | 5 (62.3) | +2 |
| multilingual-e5-large | 4 (60.7) | 7 (60.9) | +3 |
| e5-mistral-7b-instruct | 5 (60.4) | 1 (66.6) | -4 |
| multilingual-e5-base | 6 (57.9) | 9 (59.1) | +3 |
| multilingual-e5-small | 7 (56.4) | 11 (57.0) | +4 |
| multilingual-mpnet-base | 8 (53.2) | 12 (54.6) | +4 |
| LaBSE | 9 (50.5) | 14 (45.21) | +5 |
| multilingual-MiniLM-L12 | 10 (50.0) | 13 (52.5) | +3 |
| e5-large | 11 (47.7) | 6 (61.4) | -5 |
| e5-base | 12 (46.6) | 8 (60.4) | -4 |
| e5-small | 13 (45.6) | 10 (58.9) | -3 |
| mxbai-embed-large-v1 | 13 (45.6) | 2 (64.7) | -11 |

Table 3: Comparison of model performance across MTEB and SEB. Models are selected among comparable well-performing models on both benchmarks. Scores represent rank and average score on the benchmark. Rank is computed only among the selected models.

governmental, and medical fields, which are currently only partly covered or unaddressed. Current tasks predominantly feature non-fiction literature, such as encyclopedias and news, yet the rising interest in digital humanities (Su et al., 2020) suggests the inclusion of fiction, poetry, historical texts, and religious documents in future updates could be valuable. Additionally, the benchmark currently lacks some task categories, such as pair classification and document reranking.

**Future Directions:** While this work announces the release of SEB, we plan to continually expand upon the benchmark. As this work continues to develop, we invite researchers to join us in expanding the evaluation of embedding models across a broad range of languages.

**Effect of Training Data:** While we explore the influence of the learning objective on model performance, its important to acknowledge the significant role of training datasets. For example, the e5-mistral-7b-instruct model performs similarly to the much smaller multilingual-e5-large, likely due to Mistrals pre-training dataset reportedly not containing Scandinavian languages. However, the lack of transparency around the training datasets of many models means that such claims remain speculative. Future research should aim to investigate the impact of training data more thoroughly.

# 6 Conclusion

In this work, we introduced SEB, a framework that addresses the evaluation gap for the mainland Scandinavian languages. SEB encompasses 24 tasks covering ten subtasks in four task categories and spanning mainland Scandinavian languages.

We evaluate more than 50 models on SEB and show that there is still a notable gap in performance between publicly available text embedding models and their commercial counterparts, especially in retrieval contexts, as well as between monolingual and multilingual models. These findings highlight critical areas for future advancements. By open-sourcing SEB and integrating it with MTEB, we aim to encourage the development of robust Scandinavian and multilingual embedding models, inviting the research community to contribute to this evolving landscape.

## Acknowledgments and Disclosure of Funding

Part of the computation for this project was performed on the DeiC interactive HPC system via UCloud managed by the Danish consortium for Interactive HPC. The computation was performed using a 40GB NVIDIA Tesla A100 GPU.

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

Table 4: This table provides an overview, along with reference to their implementation. If a link isn't provided, it denotes the name on Huggingface.

| Name | Reference |
|---|---|
| *Self-Supervised Models* | |
| dfm-encoder-large
    + SimCSE | `danish-foundation-models/encoder-large-v1`
`Anonymized` |
| xlm-roberta-large | `FacebookAI/xlm-roberta-large` |
| nb-bert-large | `NbAiLab/nb-bert-large` |
| nb-bert-base | `NbAiLab/nb-bert-base` |
| bert-base-swedish | `KBLab/bert-base-swedish-cased` |
| fasttext-cc-da | `https://fasttext.cc/docs/en/crawl-vectors.html` |
| fasttext-cc-nn | `https://fasttext.cc/docs/en/crawl-vectors.html` |
| fasttext-cc-nb | `https://fasttext.cc/docs/en/crawl-vectors.html` |
| fasttext-cc-sv | `https://fasttext.cc/docs/en/crawl-vectors.html` |
| *Supervised Models* | |
| multilingual-MiniLM-L12 | `sentence-transformers/paraphrase-multilingual-MiniLM-L12-v2` |
| multilingual-mpnet-base | `sentence-transformers/paraphrase-multilingual-mpnet-base-v2` |
| labSE | `sentence-transformers/LaBSE` |
| sentence-bert-swedish | `KBLab/sentence-bert-swedish-cased` |
| e5-mistral-7b-instruct | `intfloat/e5-mistral-7b-instruct` |
| multilingual-e5-large | `intfloat/multilingual-e5-large` |
| multilingual-e5-base | `intfloat/multilingual-e5-base` |
| multilingual-e5-small | `intfloat/multilingual-e5-small` |
| translate-e5-large | Custom Implementation |
| sonar-dan | `facebook/SONAR` |
| sonar-nob | `facebook/SONAR` |
| sonar-nno | `facebook/SONAR` |
| sonar-swe | `facebook/SONAR` |
| *Embedding APIs* | |
| text-embedding-3-large | `https://openai.com/blog/new-embedding-models-and-api-updates` |
| text-embedding-3-small | `https://openai.com/blog/new-embedding-models-and-api-updates` |
| embed-multilingual-v3.0 | `https://txt.cohere.com/introducing-embed-v3/` |

# A  Appendix

## A.1  Models

The Table 4 reference to each of the model's names denoted in the main paper, which have been shortened for clarity.

## A.2  Domains Generalization

We see the performance across domains in Table 5. These results are generally in accordance with the results across tasks; showing that the e5 models along with the commercial APIs constitute the most competitive models.

## A.3  Dataset Embeddings

To examine the spread and similarity of our datasets, we explore their similarity in the embedding space in Figure 4. To do so, we use one of the best-performing embedding models, embed-multilingual-v3.0. We see that certain datasets occupy distinct clusters, indicating that evaluations without these datasets would likely bias the model evaluation. Notably, we additionally see that the existing (translated) datasets within MTEB (Massive Intent and Massive Scenario) cover only a small subsection of the embedding space.

Table 5: Performance across domains in SEB. The best score in each model category is highlighted in bold. We only include domains that contain at least two datasets. Additional model evaluation can be found on the public Dashboard.

| | Avg. | Fiction | Legal | News | N.-fiction | Review | Social | Spoken | Web | Wiki |
|---|---|---|---|---|---|---|---|---|---|---|
| Num. Datasets (→) | 24 | 4 | 2 | 6 | 13 | 2 | 6 | 4 | 3 | 6 |
| *Self-Supervised Models* | | | | | | | | | | |
| dfm-encoder-large | 41.4 | 44.5 | 69.7 | 31.4 | 33.6 | 56.8 | 42.3 | 57.0 | 29.4 | 31.0 |
|   + SimCSE | **46.6** | **46.4** | 72.0 | **40.5** | **42.7** | 58.7 | **41.2** | 60.7 | **39.3** | 37.3 |
| xlm-roberta-large | 35.3 | 41.5 | 41.3 | 24.9 | 25.3 | 55.9 | 36.2 | 54.4 | 24.4 | 26.5 |
| nb-bert-large | 46.0 | 44.0 | **73.2** | 38.7 | 42.6 | **61.6** | 36.1 | **61.7** | 30.5 | **39.9** |
| nb-bert-base | 42.1 | 42.6 | 71.8 | 28.7 | 35.1 | 57.6 | 38.4 | 58.7 | 29.0 | 35.0 |
| bert-base-swedish | 35.2 | 38.6 | 56.4 | 24.9 | 29.9 | 56.9 | 29.8 | 49.7 | 27.3 | 25.0 |
| fasttext-cc-da | 37.3 | 39.5 | 64.3 | 28.4 | 34.0 | 49.9 | 33.2 | 45.6 | 26.0 | 33.9 |
| fasttext-cc-nn | 35.8 | 38.1 | 64.2 | 24.8 | 33.6 | 47.5 | 29.2 | 43.2 | 24.0 | 35.5 |
| fasttext-cc-nb | 37.5 | 39.0 | 63.5 | 26.8 | 34.4 | 49.8 | 32.0 | 46.1 | 25.4 | 36.5 |
| fasttext-cc-sv | 36.0 | 38.3 | 62.7 | 28.0 | 33.3 | 50.9 | 30.1 | 45.8 | 26.6 | 29.3 |
| *Supervised Models* | | | | | | | | | | |
| multilingual-MiniLM-L12 | 50.0 | 43.5 | 68.4 | 43.9 | 49.1 | 59.9 | 45.4 | 57.6 | 43.6 | 41.2 |
| multilingual-mpnet-base | 53.2 | 44.2 | 72.8 | 47.3 | 52.4 | 64.7 | 49.0 | 59.7 | 45.6 | 43.3 |
| labSE | 50.5 | 49.0 | 71.3 | 41.9 | 48.5 | 61.9 | 48.5 | 57.7 | 48.6 | 44.6 |
| sentence-bert-swedish | 46.6 | 40.4 | 59.9 | 44.1 | 47.1 | 57.5 | 36.8 | 53.9 | 44.9 | 36.1 |
| e5-mistral-7b-instruct | 60.4 | **53.7** | **77.6** | 52.3 | 58.0 | 70.1 | **58.0** | **64.5** | **62.1** | **57.0** |
| multilingual-e5-large | **60.7** | 48.1 | 76.1 | **54.5** | **58.9** | **73.5** | 54.9 | 62.0 | 54.9 | 55.7 |
| multilingual-e5-base | 57.9 | 48.5 | 74.9 | 50.4 | 56.2 | 69.6 | 52.6 | 59.7 | 54.3 | 54.8 |
| multilingual-e5-small | 56.4 | 49.0 | 72.3 | 50.8 | 55.4 | 65.9 | 51.1 | 57.8 | 54.8 | 53.4 |
| translate-e5-large | 47.7 | 43.2 | 69.4 | 36.8 | 43.7 | 68.1 | 46.5 | 55.5 | 40.1 | 37.8 |
| sonar-dan | 43.4 | 50.2 | 73.5 | 31.0 | 35.7 | 59.1 | 49.2 | 55.5 | 43.0 | 33.1 |
| sonar-nob | 41.5 | 45.2 | 70.1 | 28.0 | 34.1 | 57.9 | 43.8 | 55.6 | 35.8 | 31.0 |
| sonar-nno | 41.5 | 46.5 | 71.3 | 28.4 | 33.9 | 58.5 | 44.8 | 56.0 | 37.7 | 30.0 |
| sonar-swe | 42.8 | 50.5 | 73.2 | 30.9 | 35.9 | 58.2 | 47.0 | 55.0 | 44.1 | 33.5 |
| *Embedding APIs* | | | | | | | | | | |
| text-embedding-3-large | **65.0** | **50.5** | 76.1 | 56.1 | **64.1** | 72.7 | **59.0** | **64.4** | **61.0** | **65.5** |
| text-embedding-3-small | 61.0 | 50.2 | 75.9 | 54.0 | 61.2 | 66.6 | 55.3 | 61.2 | 58.1 | 60.7 |
| embed-multilingual-v3.0 | 64.1 | 49.2 | **76.6** | **56.2** | 63.5 | **75.2** | 57.1 | 63.3 | 57.9 | 63.6 |

## A.4 Evaluation and Metrics

This section briefly presents the tasks, their evaluation, and their metric. However, we utilize a similar implementation as MTEB to keep results comparable. Thus we refer to the original work for a more detailed introduction. We do, however, make one notable difference, that is, we allow the models to embed the tasks differently depending on the task, this is especially relevant for the e5 models, embed-multilingual-v3.0 and prompt-based models such as e5-mistral-7b-instruct.

**Classification:** Using the embedding model a train and a test set are embedded. Using the embedding training set a logistic classifier is trained using a maximum of 100 iterations. The model is then tested on the test set. This approach is repeated 10 times and metrics are calcuated for each set and aggregated. The training sets for these repeats are obtaining by sampling from the training set (with replacement) 16 examples from each label. The mean accuracy is reported as the main metric. Other metrics reported include the F1-score and measures of uncertainty of metrics (standard error).

**Bitext Mining:** The dataset consists of matching pairs of sentences, and the goal is to find the match. All matching pairs of sentences are embedded using the embedding model. Afterward, the closest match is found using cosine similarity. F1 is reported as the main metric.

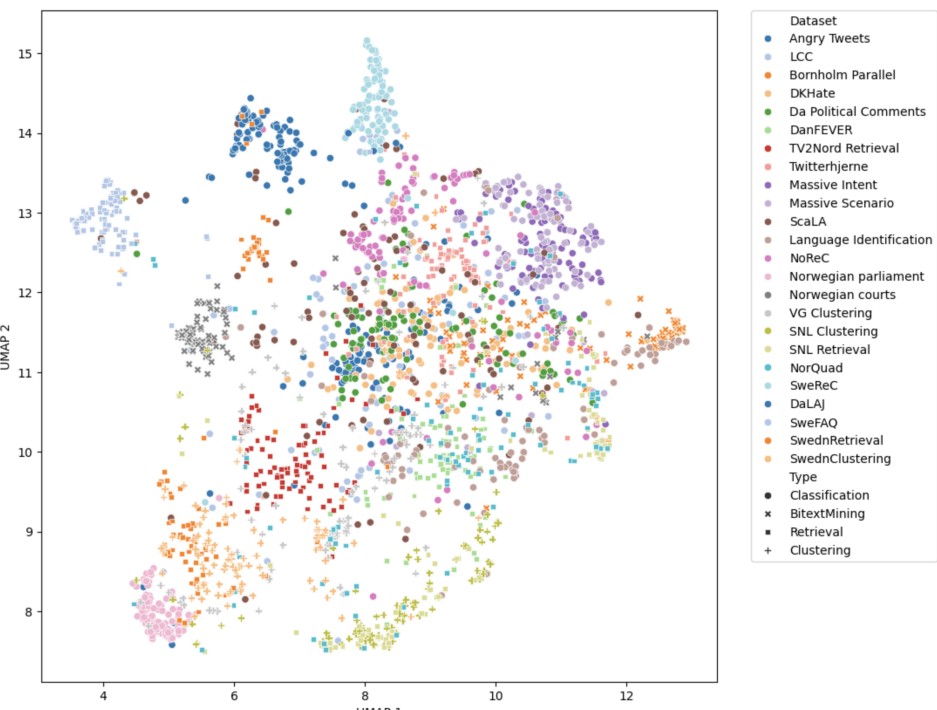

Figure 4: The embeddings of 100 randomly sampled documents from each task, embedded using embed-multilingual-v3.0 and projected using a UMAP projection. The project uses the cosine metrics but otherwise default parameter values.

**Clustering** The dataset consists of documents attached with a label, e.g., a denoted category such as "sports." The goal is the correctly cluster the documents into similar clusters as the labels. All documents are embedded, and a mini-batch k-means model (batch size 32 and k equal to the number of unique labels in the dataset) is trained on the embeddings. The V measure is used as is reported as the main metric, as the permutation of labels does not affect the score.

**Retrieval:** The dataset consists of a corpus, queries as well as a mapping between the queries and their relevant documents. The goal is to retrieve these relevant documents. Both queries and documents are embedded using the model. We allow these to be embedded differently depending on the model. For each query, the corpus documents are ranked using a similarity score, and nDCG@10 is reported as the main metric.

## A.5 Datasets Construction

This section briefly describes the construction of the tasks.

**Classification:** As all the classification datasets are derived from existing datasets, no additional processing is done to these except to limit the size of excessively large datasets.

**Bitext Mining:** Similar to the classification, these datasets already existed as paired datasets. With the Norwegian Courts being extracted from OPUS (Tiedemann, 2012) and Bornholm Parallel being derived from (Derczynski and Kjeldsen, 2019).

**Clustering:** For clustering, we construct the dataset based on existing datasets of news or encyclopedic corpora (Navjord and Korsvik, 2023; Berdicevskis et al., 2023) using their attached categories. The SNL and VG datasets (Navjord and Korsvik, 2023) contain a hierarchy of labels; here, we subjectively chose a meaning level and validated that it led to a meaningful separation in performance – using either too many or too few levels would to either 1-2 clusters or clusters consisting of only 2-3 documents.

Similar to the classification, these datasets already existed as paired datasets. With the Norwegian Courts being extracted from OPUS (Tiedemann, 2012) and Bornholm Parallel being derived from (Derczynski and Kjeldsen, 2019).

**Retrieval:** For the construction of the retrieval datasets, we used either question and answer datasets (e.g., NorQuad (Ivanova et al., 2023)) or news summarization datasets (e.g., (Berdicevskis et al., 2023)). For the question and answer we specified the questions as queries and the answers as the corpus. A correct answer was considered to be a properly retrieved document. For the summaries, we considered the headline as the query and both the summaries and the articles as the corpus. Matching summaries and articles were considered properly retrieved documents.

## A.6 Datasets Statistics

Table 6 contains an overview of each of the datasets in SEB, including a short description, descriptive statics, task formalization, and domains as defined by (Nivre et al., 2017).

| Dataset | Description | Main Score | Langs | Type | Domains | N. Docs | Avg. Length |
|---------|-------------|------------|-------|------|---------|---------|-------------|
| Angry Tweets Pauli et al. (2021) | A sentiment dataset with 3 classes (positiv, negativ, neutral) for Danish tweets | Accuracy | da | Classification | social | 1047 | 156.15 (82.02) |
| Bornholm Parallel Derczynski and Kjeldsen (2019) | Danish Bornholmsk Parallel Corpus. Bornholmsk is a Danish dialect spoken on the island of Bornholm, Denmark. | F1 | da, da-bornholm | BitextMining | poetry, wiki, fiction, web, social | 1000 | 44.36 (41.22) |
| DKHate Sigurbergsson and Derczynski (2020) | Danish Tweets annotated for Hate Speech either being Offensive or not | Accuracy | da | Classification | social | 329 | 88.18 (68.30) |
| Da Political Comments | A dataset of Danish political comments rated for sentiment | Accuracy | da | Classification | social | 7206 | 69.60 (62.85) |
| DaLAJ Berdicevskis et al. (2023) | A Swedish dataset for linguistic acceptability. Available as a part of Superlim | Accuracy | sv | Classification | fiction, non-fiction | 888 | 120.77 (67.95) |
| DanFEVER Nørregaard and Derczynski (2021) | A Danish dataset intended for misinformation research. It follows the same format as the English FEVER dataset. | NDCG@10 | da | Retrieval | wiki, non-fiction | 8897 | 124.84 (168.53) |
| LCC Nielsen (2016) | The Leipzig corpora collection, annotated for sentiment | Accuracy | da | Classification | legal, web, news, social, fiction, non-fiction, academic, government | 150 | 118.73 (57.82) |
| Language Identification Haas and Derczynski (2021) | A dataset for Nordic language identification. | Accuracy | da, sv, nb, nn, is, fo | Classification | wiki | 3000 | 78.23 (48.54) |
| Massive Intent FitzGerald et al. (2023) | The intent task within MASSIVE corpus translated for Scandinavian languages | Accuracy | da, nb, sv | Classification | spoken | 15021 | 34.65 (16.99) |
| Massive Scenario FitzGerald et al. (2023) | The scenario task within MASSIVE corpus translated for Scandinavian languages | Accuracy | da, nb, sv | Classification | spoken | 15021 | 34.65 (16.99) |

| Dataset | Description | Main Score | Langs | Type | Domains | N. Docs | Avg. Length |
|---|---|---|---|---|---|---|---|
| NoReC Velldal et al. (2018) | A Norwegian dataset for sentiment classification on review | Accuracy | nb | Classification | reviews | 2048 | 89.62 (61.21) |
| NorQuad Ivanova et al. (2023) | Human-created question for Norwegian Wikipedia passages. | NDCG@10 | nb | Retrieval | non-fiction, wiki | 2602 | 502.19 (875.23) |
| Norwegian courts Tiedemann (2012) | Nynorsk and Bokmål parallel corpus from Norwegian courts. | F1 | nb, nn | BitextMining | legal, non-fiction | 456 | 82.11 (49.48) |
| Norwegian parliament | Norwegian parliament speeches annotated with the party of the speaker ('Sosialistisk Venstreparti' vs 'Fremskrittspartiet') | Accuracy | nb | Classification | spoken | 2400 | 1897.51 (1988.62) |
| SNL Clustering Navjord and Korsvik (2023) | Webscrabed articles from the Norwegian lexicon 'Det Store Norske Leksikon'. Uses article's categories as clusters. | V measure | nb | Clustering | non-fiction, wiki | 2048 | 1101.30 (2168.35) |
| SNL Retrieval Navjord and Korsvik (2023) | Webscrabed articles and ingresses from the Norwegian lexicon 'Det Store Norske Leksikon'. | NDCG@10 | nb | Retrieval | non-fiction, wiki | 2600 | 1001.43 (2537.83) |
| ScaLA Nielsen (2023) | A linguistic acceptability task for Danish, Norwegian Bokmål Norwegian Nynorsk and Swedish. | Accuracy | da, nb, sv, nn | Classification | fiction, news, non-fiction, spoken, blog | 8192 | 102.45 (55.49) |
| SweFAQ Berdicevskis et al. (2023) | A Swedish QA dataset derived from FAQ | NDCG@10 | sv | Retrieval | non-fiction, web | 1024 | 195.44 (209.33) |
| SweReC Nielsen (2023) | A Swedish dataset for sentiment classification on review | Accuracy | sv | Classification | reviews | 2048 | 318.83 (499.57) |
| SwednClustering Berdicevskis et al. (2023) | News articles from the Swedish newspaper Dagens Nyheter (DN) collected during the years 2000–2020. Uses the category labels as clusters. | V measure | sv | Clustering | non-fiction, news | 2048 | 1619.71 (2220.36) |
| SwednRetrieval Berdicevskis et al. (2023) | News articles from the Swedish newspaper Dagens Nyheter (DN) collected during the years 2000–2020. | NDCG@10 | sv | Retrieval | non-fiction, news | 3070 | 1946.35 (3071.98) |

| Dataset | Description | Main Score | Langs | Type | Domains | N. Docs | Avg. Length |
|---------|-------------|-----------|-------|------|---------|---------|-------------|
| TV2Nord Retrieval | News Article and corresponding summaries extracted from the Danish newspaper TV2 Nord. | NDCG@10 | da | Retrieval | news, non-fiction | 4096 | 784.11 (982.97) |
| Twitterhjerne Holm (2024) | Danish question asked on Twitter with the Hashtag #Twitterhjerne ('Twitter brain') and their corresponding answer. | NDCG@10 | da | Retrieval | social | 340 | 138.23 (82.41) |
| VG Clustering Navjord and Korsvik (2023) | Articles and their classes (e.g. sports) from VG news articles extracted from Norsk Aviskorpus. | V measure | nb | Clustering | non-fiction, news | 2048 | 1009.65 (1597.60) |

Table 6: Tasks available in SEB. The average length is specified in characters. Values in parentheses denote the standard deviation.

## A.7  Long-term Availability and Stability

All of the datasets used for the Scandinavian Embedding Benchmark are publicly available on Huggingface repositories. To avoid duplicating metadata, we refer to and download from existing repositories, however, to ensure stability, we refer to a specific revision used. This allows us to update the benchmark datasets if annotations are corrected or faulty entries are removed. Additionally, we keep a copy of all datasets in case datasets are removed from the Huggingface Hub such that they can be re-uploaded. The permissible licenses of the datasets ensure that this is a viable option.

## A.8  Results per Task

In the following figure, we see an overview of all of the results of the benchmark for each task for the selected models. To get an up-to-date overview, check out the online dashboard.

