# OpenReview forum: "The Scandinavian Embedding Benchmarks: Comprehensive Assessment of Multilingual and Monolingual Text Embedding"
_NeurIPS.cc/2024/Datasets_and_Benchmarks_Track — NeurIPS 2024 Track Datasets and Benchmarks Poster_

### Official Review · Reviewer_YWqP · 2024-07-07
**The Scandinavian Embedding Benchmarks**

**Rating:** 7
**Confidence:** 4

**Review:**

### Quality
The quality of the benchmark based on the proposed purpose is good but could be slightly higher. Some areas are quite obvious that they could be extended to all languages. For example, only the Norwegian parliament is included, even though the Danish and Swedish parliaments could probably be easily added (e.g. see the ParlaMint collaboration). Also, most parts of the benchmark have already been published. I count to three tasks that have not already been published. Similarly, the authors mention public reports and other data on the welfare state can be added in the future. This could have been added already. Although, one needs to draw a line somewhere.

### Originality
To the best of my knowledge, the proposed embedding is not very original (see, e.g. MTEB), but an embedding benchmark for the Scandinavian countries does not exist. That the paper builds upon the ideas of MTEB can be seen as less novel. However, I would rather argue that this “lack of novelty” is good in this setting since it simplifies the benchmarking of previous work and can be seen as an extension of the MTEB benchmark, currently lacking in Scandinavian languages. I think the focus on comparability is very good here. More papers and dataset contributions should be made like this to facilitate research rather than be isolated datasets/benchmarks.

### Significance
The impact is not necessarily very large since the Scandinavian language is a small language family (maybe 20 million native speakers). However, it is still a good contribution, and I think it is important to extend these benchmarks outside the standard English setting. The proposed benchmark is also using data that is not simply automatically translated. The fact that the authors simplify the combination of SEB with MTEB is very good for further simplifying research on embedding quality.

For clarity, strengths and weaknesses, see below.

**Strengths:**

The main strength of the paper is the contribution to larger sentence/document embedding benchmarking literature and to enable additional languages for evaluation purposes. Segment embeddings are becoming increasingly important in many settings, so further and more general evaluations are needed.

**Additional Feedback:**

-

**Clarity:**

The paper is clearly written and easy to follow. A small note is that the subsection should probably use capital letters since it is a proper noun (just as Table and Figure).

**Correctness:**

I think the claims made in the submission are correct and seem to be constructed in a sound way.

**Documentation:**

The appendices contain additional information on how the benchwork was created and assembled. They also include a URL to the benchmark with additional (but minimal) documentation. The main information can be found in the paper's appendices.

**Ethics:**

No.

**Limitations:**

The authors point to the current scope limitations and mention the limitation languages. I think the authors need to be more explicit that Icelandic and Faroese are currently only included in the corpus in one place.

The authors also write that they have bootstrapped the results, but judging how this has been done is difficult. I would suggest the paper include how this bootstrap was conducted to judge if it was properly done. It is also unclear if the confidence interval is used? Also, why is the confidence interval not included for the individual tasks? I think that is needed for the benchwork to be suitable for comparisons/benchmarking.

The authors use both F1 score and accuracy. Why not use F1 throughout?

In A.4, the authors seem to do 10-fold cross-validation. Is this the case? Also, why is the logistic classifier run for 100 iterations? Is this due to the problem that for some embedding models, there are more embedding dimensions than observations? This is generally unclear to me since, for some datasets, this would be the case, and then, e.g., maximum likelihood estimation would not be feasible. Or do I miss something?

Also, evaluating clustering using supervised information (categories) is slightly strange. Why should categories be the way the clustering is evaluated? That does not make sense to me. I also think this problem is shown in the results. All embeddings are more or less equally bad, but the best one is a smaller version, which is hard to explain. Why do we see this result? Is that model better? Probably not. This indicates that the task might not have been constructed sensibly. I suggest the authors treat this as a classification task instead since this makes more sense.

**Opportunities For Improvement:**

The Scandinavian languages are quite a small group of languages, and they are not very different from English. So, even if the paper contributes to non-English languages, it might still be hard to use for generalisation performance in languages different from English.

Two additional languages are also missing in the Scandinavian family: Faroese and Icelandic. These languages are part of the language identification tasks in the paper, so I think it could have been included in some way. Faroese is a low-resource language, which would be an important reason for inclusion. It is mentioned in the paper that it concerns the mainland Scandinavian languages, which probably should be clarified more in the paper.

See
Hansen, Kirsti. (2015-05-27). FTS - Faroese text collection [Data set]. Språkbanken Text. https://doi.org/10.23695/r8td-xw20
for some information and examples on Faroese.

**Relation To Prior Work:**

The authors relate the benchmark to the MTEB, which is the main previous work of the proposed embedding benchmark.

**Summary And Contributions:**

The paper introduces the Scandinavian embedding benchmark (SEB), a sentence/document embedding task set for the three larger Scandinavian languages. The SEB is closely aligned with the Massive Text Embedding Benchmark (MTEB) and extends the MTEB with the (mainland) Scandinavian languages in multiple tasks and with multiple (cached) models.

---

> ### Author Response · Authors · 2024-08-19
>
> We thank the reviewer for the detailed review and respond to specific points below:
>
>
> > Also, most parts of the benchmark have already been published [...]
>
>
> Benchmark papers (e.g. glue (Wang et al., 2018)) do indeed often utilize already published datasets. This has the benefit of providing a higher degree of dataset quality as errors will likely have been found or corrected. Thus we do not consider re-using existing datasets a limitation, but rather were there enough diverse and high-quality datasets available, we would likely have avoided including new data sources.
>
>
> > Similarly, the authors mention public reports and other data on the welfare state can be added in the future. This could have been added already. Although, one needs to draw a line somewhere.
>
>
> We are currently processing dataset agreements with some partners, but there is no guarantee that these will result in a publishable dataset nor when exactly this will happen. We are however committed to continually developing the benchmark.
>
>
> > [...] For example, only the Norwegian parliament is included, even though the Danish and Swedish parliaments could probably be easily added (e.g. see the ParlaMint collaboration).
>
>
> We thank the reviewer for making us aware of this dataset. We have created an issue and are working on adding it to the benchmark. We are actively looking for other ways to extend and improve the quality of our benchmark, please let us know if you have additional suggestions.
>
>
>
>
> > The Scandinavian languages are quite a small group of languages, and they are not very different from English. So, even if the paper contributes to non-English languages, it might still be hard to use for generalisation performance in languages different from English.
>
>
> The Mainland Scandinavian languages are indeed also Germanic languages similar to English. Thus one might expect that conclusions made using MTEB would generalize to SEB, however, as we see the comparison between scores on MTEB and SEB this is not the case (see Table X in response to reviewer 1). If we already see this in low-mid resources languages within the same language family it is likely much more pronounced in more distant language families.
>
>
> To be clear we do not intend to claim that our benchmark measures performance in languages outside of the Mainland Scandinavian languages.
>
>
> > [...] Two additional languages are also missing in the Scandinavian family: Faroese and Icelandic [...]
> > [...] I think the authors need to be more explicit that Icelandic and Faroese are currently only included in the corpus in one place.
>
>
> Faroese and Icelandic could indeed be added to SEB. We based these limitations on previous research on Scandinavian Benchmarks (Nielsen, 2023), which show “that there is substantial cross-lingual transfer among the Mainland Scandinavian languages (Danish, Swedish and Norwegian), with limited cross-lingual transfer between the group of Mainland Scandinavian languages and the group of Insular Scandinavian languages (Icelandic and Faroese)”. Thus aggregating across Mainland and Insular Scandinavian Languages was deemed inappropriate. We will make this argumentation clear within the paper:
>
>
> “Please note that the Insular Scandinavian languages, Icelandic and Faroese, are not included as previous research (Nielsen, 2023) has demonstrated only a limited degree of cross-lingual transfer between the Mainland and Insular Scandinavian languages.”
>
>
> However, we do agree with the reviewer that future efforts should seek to broaden the coverage of MTEB to a much wider set of (low-resource) languages. Such a process is already underway as a part of the multilingual MTEB (MMTEB) open collaboration.
>
>
> > [...] I would suggest the paper include how this bootstrap was conducted to judge if it was properly done.
>
>
> The 95% confidence interval provided in Table 2 is obtained by bootstrapping based on task performance (sample N tasks with replacement, where N is the total number of tasks) to minimize the impact of any single task. We use 100 repetitions as this results in a robust estimate. We also considered using bootstrapping based on each task score as the reviewer suggests, but found that this would require dropping compatibility with MTEB or major refactors of MTEB. We thus agree with the authors that uncertainty around these estimates would be ideal and are happy to say that progress has begun in adding these uncertainty metrics to MTEB and SEB jointly.
>
>
> > The authors use both F1 score and accuracy. Why not use F1 throughout?
>
>
> We compute F1 and Accuracy for both Bitext Mining and Classification tasks. We additionally compute precision and recall as well as the average precision for classification. The choice of metric is to keep our results comparable with MTEB. However, the result object contains all metrics.
>
>
> > A small note is that the subsection should probably use capital letters [...]
>
>
> Thanks for the correction, we have made the appropriate changes.

---

> > ### Comment · Reviewer_YWqP · 2024-08-31
> >
> > Thank you for your extensive response. I'm willing to improve my score based on the additions suggested by the authors.

---

> > > ### Author Response · Authors · 2024-09-01
> > >
> > > Thank you for appreciating the effort we put in. I believe it is possible to edit the score in the original review to reflect the new score.

---

> ### Author Response · Authors · 2024-08-19
> **References**
>
> - Alex Wang, Amanpreet Singh, Julian Michael, Felix Hill, Omer Levy, and Samuel Bowman. 2018. GLUE: A Multi-Task Benchmark and Analysis Platform for Natural Language Understanding. In Proceedings of the 2018 EMNLP Workshop BlackboxNLP: Analyzing and Interpreting Neural Networks for NLP, pages 353–355, Brussels, Belgium. Association for Computational Linguistics.
>
> - Dan Nielsen. 2023. ScandEval: A Benchmark for Scandinavian Natural Language Processing. In Proceedings of the 24th Nordic Conference on Computational Linguistics (NoDaLiDa), pages 185–201, Tórshavn, Faroe Islands. University of Tartu Library.

---

### Official Review · Reviewer_TXiX · 2024-07-20
**SEB follows up and complements the most widely used embedding benchmark.**

**Rating:** 9
**Confidence:** 4
**Correctness:** The experimental setup seems thorough…

**Review:**

The benchmark is well designed to gauge the quality of embeddings in Scandinavian languages: it is multi-task, multilingual (among the Scandinavian languages), cross lingual, it considers cultural aspects and it is designed to improve on the reproducibility constraints of previous benchmarks. The new benchmark is focused on Scandinavian languages, but it correctly addresses both the expansion to other languages by having monolingual and cross-lingual tasks. Including languages from the same family is useful to study regional and cultural nuances.

**Strengths:**

The fair and constructive criticism of MTEB is instructive and welcome. The discussion provides good insights about model size, embedding size, training style, etc. that is valuable for the analysis of dense retrieval in general.

**Additional Feedback:**

Glad to see this MTEB extension!

**Clarity:**

The paper is clearly written and well-structured. The figures are illustrative and the details provided about the data and tasks are clear and sufficient.

**Documentation:**

I couldn’t find in the documentation further information about the model registry or its mechanics. Also, it was a bit hard to understand the way the “translate and embed” setup.

**Ethics:**

No questions or concerns.

**Limitations:**

The authors already list a valid list of limitations in the papers. No additional ones noted.

**Opportunities For Improvement:**

The translate-and-embed approach may benefit from using LLMs for the translation of both the queries and the candidates followed by retrieval with English-only models. Even though it may be cumbersome, evaluating the translations with the models at the top of the MTEB leaderboard may provide some insight into how strong those models are in this benchmark.

If available, it would be useful to add the MTEB results of the multilingual models tested. That would help to put the MTEB results in context as many of the models topping the leaderboard may have been built with the specific purpose to rank high in MTEB. Showing that there are multilingual models that are versatile enough to rank high in SEB should give more credence to their MTEB results, because they are not tailor-made for that specific benchmark.

**Relation To Prior Work:**

The paper correctly builds on top of previous work like BEIR and MIRACL, which led to MTEB. SEB can be seen as an extension to MTEB that could be the seed to expand it to other languages. It builds on MTEB by addressing its limitations while remaining compatible with it.

**Summary And Contributions:**

The authors introduce SEB: a comprehensive evaluation dataset of Scandinavian languages including 24 evaluation tasks. The benchmark includes text in 4 languages plus a dialect. This is a valuable addition to complement English-only benchmarks like MTEB. The authors do thorough experimentation with existing retrieval multilingual models showing that monolingual and smaller (Scandinavian) models do not perform as well in this benchmark, but the large LLM-based multilingual models do.

Importantly, this work sets the basis for a multilingual expansion of MTEB as well as improving the reproducibility of computing these results.

---

> ### Author Response · Authors · 2024-08-19
>
> We sincerely thank the reviewer for their positive feedback and valuable suggestions.
>
>
> > If available, it would be useful to add the MTEB results of the multilingual models tested. That would help to put the MTEB results in context as many of the models topping the leaderboard may have been built with the specific purpose to rank high in MTEB. Showing that there are multilingual models that are versatile enough to rank high in SEB should give more credence to their MTEB results, because they are not tailor-made for that specific benchmark.
>
>
> We fully agree with your suggestion. In response to this, we have included a comparison of MTEB and SEB within the results. We have additionally attached the table in our response to Reviewer 1 and will ensure this comparison is featured in the published paper.
>
>
> > The translate-and-embed approach may benefit from using LLMs for the translation of both the queries and the candidates followed by retrieval with English-only models. Even though it may be cumbersome, evaluating the translations with the models at the top of the MTEB leaderboard may provide some insight into how strong those models are in this benchmark.
>
>
> We acknowledge that the translate-and-embed baseline could indeed benefit from higher quality translations and embedding models. However, as it stands, this approach underperforms compared to simpler alternatives, such as multilingual-e5-small. Given the increased complexity of the pipeline, any performance improvement would need to be significant to justify the additional complexity.
>
>
> > I couldn't find in the documentation further information about the model registry or its mechanics. Also, it was a bit hard to understand the way the “translate and embed” setup.
>
> In the associated GitHub you can find the model registry within `src/seb/registrered_models` here you can find the implementation of all of the models including the “translate and embed” which is present in the `translate_e5_models.py` file.
>
> When using the codebase you can retrieve models from the registry using the command `model = seb.get_model("translate-e5-large")`. We also include an example of this within our ‘getting started’ guide.
>
>
> The setup for the “translate and embed” model is as follows: Inputs are translated using the M2M100 model (chosen to match the language of the text). Then the text is encoded using the e5-large. We supply the model with the correct language, though it would probably have been more appropriate to include language detection. However, as this would result in accumulating errors and the models already perform poorly we deemed it unnecessary as in many applications the language is known.

---

### Official Review · Reviewer_5G2G · 2024-07-26
**A text embedding benchmark for Scandinavian languages**

**Rating:** 5
**Confidence:** 2
**Clarity:** The paper is well written.

**Review:**

The paper presents a well-structured benchmark with clear task definitions, dataset descriptions, and evaluation metrics. The method used  for the benchmark construction is not novel, though.

**Strengths:**

* The construction of SEB and its release to the public will facilitate research on text embeddings for Scandinavian languages
* Detailed analysis over 26 models on SEB, showing some insights between self-supervised and supervised models.

**Additional Feedback:**

would be great to add paper links to each model in Table 2.

**Correctness:**

* The dataset construction and evaluation sounds reasonable to me.
* While the authors claimed a comprehensive assessment of monolingual and multilingual text embeddings, the analysis presented in the paper is not that "comprenehsive"

**Documentation:**

The authors released the dataset, and provided description about how they constructed it.

**Ethics:**

I didn't find serious issues.

**Limitations:**

The author discussed limitations.

**Opportunities For Improvement:**

* While the paper presents overall performance across task categories, a deeper analysis of model behavior on individual tasks is missing. This limits the understanding of model strengths and weaknesses.
* It would be great to list the performance of different models on MTEB so as to illustrate how performance on MTEB differs from SEB.
* The performance difference across different models may not be attributed to the learning objectives alone as the used pretraining data also differs greatly.

**Relation To Prior Work:**

The authors discussed relation to previous work adequately

**Summary And Contributions:**

This paper introduces Scandinavian Embedding Benchmark (SEB), a comprehensive framework for evaluating the quality of text embeddings in Scandinavian languages. It covers 24 tasks, 10 subtasks and 4 task categories, spanning 12 distinct domains, forming a complement to existing benchmarks like MTEB. The authors evaluated over 26 models, analyzed their performance, and explored the trade-off between performance and efficiency.

---

> ### Author Response · Authors · 2024-08-19
> **Response to Reviewer 5G2G**
>
> We thank the reviewer for their feedback and respond to specific points below:
> > While the authors claimed a comprehensive assessment of monolingual and multilingual text embeddings, the analysis presented in the paper is not that "comprehensive."
>
> We acknowledge that while our benchmarks represent a significant advancement for low- to mid-resource Scandinavian languages, it may be misleading to describe the assessment as fully comprehensive. Therefore, we have rephrased the title to: "The Scandinavian Embedding Benchmarks: Evaluating Multilingual and Monolingual Text Embeddings for Scandinavian Languages." Additionally, we have made corresponding adjustments throughout the text.
>
>
> > It would be great to list the performance of different models on MTEB to illustrate how performance on MTEB differs from SEB.
>
>
> We agree that a comparison between SEB and MTEB is valuable and have included the following table:
>
>
> | Model                   | SEB       | MTEB       | Difference |
> | ----------------------- | --------- | ---------- | ---------- |
> | text-embedding-3-large  | 1 (65.0)  | 3 (64.6)   | +2         |
> | embed-multilingual-v3.0 | 2 (64.1)  | 4 (64.0)   | +2         |
> | text-embedding-3-small  | 3 (61.0)  | 5 (62.3)   | +2         |
> | multilingual-e5-large   | 4 (60.7)  | 7 (60.9)   | +3         |
> | e5-mistral-7b-instruct  | 5 (60.4)  | 1 (66.6)   | -4         |
> | multilingual-e5-base    | 6 (57.9)  | 9 (59.1)   | +3         |
> | multilingual-e5-small   | 7 (56.4)  | 11 (57.0)  | +4         |
> | multilingual-mpnet-base | 8 (53.2)  | 12 (54.6)  | +4         |
> | LaBSE                   | 9 (50.5)  | 14 (45.21) | +5         |
> | multilingual-MiniLM-L12 | 10 (50.0) | 13 (52.5)  | +3         |
> | e5-large                | 11 (47.7) | 6 (61.4)   | -5         |
> | e5-base                 | 12 (46.6) | 8 (60.4)   | -4         |
> | e5-small                | 13 (45.6) | 10 (58.9)  | -3         |
> | mxbai-embed-large-v1    | 13 (45.6) | 2 (64.7)   | -11        |
>
>
> Table X: Comparison of model performance across MTEB and SEB. Models are selected among comparable well-performing models on both benchmarks. Scores represent rank and average score on the benchmark. Rank is computed only among the selected models.
>
>
> As anticipated, English-focused models (e5-{size}) perform significantly worse on SEB, while multilingual models perform relatively better. The e5-mistral-7b-instruct model, based on the multilingual Mistral, shows notably poorer performance on non-English data, likely because the Mistral model wasn’t trained on Scandinavian languages. While English has several open-source models that perform on par with APIs, this trend is not as evident for Scandinavian languages. Among multilingual models, MTEB and SEB rankings generally align, though selecting models for Scandinavian languages based solely on MTEB results could lead to suboptimal choices (e.g., e5-mistral-7b-instruct, LaBSE).
>
>
> > While the paper presents overall performance across task categories, a deeper analysis of model behavior on individual tasks is missing. This limits the understanding of model strengths and weaknesses
>
>
> We have indeed sought to give a meaningful overview of the model performances across task categories and languages. For a more in-depth analysis we have referred the reader to the public and continually updated leaderboard. Which not only contains the information provided in Table 2, but also visuals for performance pr. task, across domains (see also Table 5) as well as continually updated visuals corresponding to Figure 3 and Table 1. It even contains aggregation across task subtypes such as those presented in Figure 1. We have included the most relevant of these within the paper and will additionally add the performance pr. task in Appendix A.8.
>
>
> > The performance difference across different models may not be attributed to the learning objectives alone as the used pretraining data also differs greatly.
>
>
> We agree that training data significantly influence model performance. Unfortunately, training data are only disclosed for a small subset of the examined models. We have added the following to the limitations section:
>
>
> "[...] While we explore the influence of the learning objective on model performance, it’s important to acknowledge the significant role of training datasets. For example, the e5-mistral-7b-instruct model performs similarly to the much smaller multilingual-e5-large, likely due to Mistral’s pre-training dataset reportedly not containing Scandinavian languages. However, the lack of transparency around the training datasets of many models means that such claims remain speculative. Future research should aim to investigate the impact of training data more thoroughly."

---

### Decision · Program_Chairs · 2024-09-26

**Decision:**

Accept (Poster)

**Comment:**

This paper introduces the Scandinavian Embedding Benchmark (SEB), a comprehensive framework for evaluating the quality of text embeddings in Scandinavian languages. SEB covers 24 tasks, 10 subtasks, and 4 task categories, spanning 12 distinct domains, serving as a valuable complement to existing benchmarks like MTEB. The authors evaluated over 26 models, analyzed their performance, and explored the trade-offs between performance and efficiency. This work lays the foundation for a multilingual expansion of MTEB and enhances the reproducibility of computational results. The reviewers all agree on the novel contribution of extending MTEB with support for Scandinavian languages. While Reviewer 5G2G raised concerns regarding the findings and the experimental comparison between SEB and MTEB, the authors have adequately addressed these issues. We encourage the authors to incorporate these revisions into their final version.